# Cardiac Disease Related to Primary Hyperthyroidism in a 20-Year-Old Mule

**DOI:** 10.3390/ani14111660

**Published:** 2024-06-01

**Authors:** Kaitlin Brown, Elizabeth Williams Louie, Toby Pinn-Woodcock, Erin Pearson, Garett B. Pearson, Jacqueline Marr, Eileen S. Hackett, Laura Rath Brown, Katharyn J. Mitchell

**Affiliations:** 1Department of Clinical Sciences, Cornell University, Ithaca, NY 14850, USA; kb652@cornell.edu (K.B.); egw43@cornell.edu (E.W.L.); tlp52@cornell.edu (T.P.-W.); eek55@cornell.edu (E.P.); gbp34@cornell.edu (G.B.P.); esh82@cornell.edu (E.S.H.); 2New Hampshire Veterinary Diagnostic Laboratory, University of New Hampshire, Durham, NH 03824, USA; jm2699@cornell.edu; 3Midstate Veterinary Services, Cortland, NY 14045, USA; dr.brown@midstatevet.com

**Keywords:** hyperthyroidism, thyroidectomy, adenocarcinoma, adenoma, ventricular hypertrophy, hypertension, murmur, equid

## Abstract

**Simple Summary:**

Historically, there have been very few reports of elevated thyroid hormone (hyperthyroidism) due to thyroid dysfunction in equid species (horses, mules, donkeys, zebras). Hyperthyroidism is a relatively common problem in cats and humans. Increased appetite, weight loss and muscle wasting, elevated heart rate and elevated blood pressure are common clinical signs in these species. In the few case reports of hyperthyroid horses, they often present with weight loss and increased appetite, but there have not been any documented cardiac changes (elevated heart rate, heart murmur, elevated blood pressure) in these patients. The mule investigated in this case report presented with a persistently elevated heart rate and systolic murmur. Cardiac imaging revealed structural changes consistent with those seen in other hyperthyroid mammals. This case report documents the diagnosis, treatment and management and provides long-term follow up of endocrine and cardiac disease in a mule with primary hyperthyroidism.

**Abstract:**

Primary hyperthyroidism is a rarely diagnosed endocrinopathy in equids and there have been no previous reports of structural and functional cardiac changes associated with hyperthyroidism in these species. This case report investigates a 20-year-old mule gelding that presented for a three-month history of thin body condition despite polyphagia, with a heart murmur and elevated free and total thyroid hormone concentrations. On presentation, physical exam revealed a body condition score of two out of nine, persistent tachycardia, pansystolic heart murmur and firm bilateral ventral proximal cervical masses. Bloodwork confirmed markedly elevated free T4, total T4 and T3 concentrations. Echocardiogram demonstrated left ventricular concentric hypertrophy with increased ventricular and atrial systolic function. Bilateral thyroidectomy was performed under standing sedation without complications. Histopathology demonstrated adenocarcinoma of the left thyroid gland and multiple adenomas with osseous metaplasia within the right thyroid. The mule was supplemented with levothyroxine sodium two weeks post-op after a thyroid panel demonstrated undetectable concentrations. Polyphagia resolved following surgery and the mule began gaining weight. Echocardiographic changes improved but did not resolve at two years post-operative. Continued bi-annual follow up and monitoring of thyroid levels was recommended. This case represents the first documentation of hemodynamically relevant cardiac remodeling in an equid associated with primary hyperthyroidism.

## 1. Introduction

Hyperthyroidism in horses has been documented [1,2,3,4,5,6,7], but it is considered a rare endocrinopathy of equids. Previously diagnosed cases have been caused by unilateral functional tumors of the thyroid, either adenomas or adenocarcinomas. Thyroid tumors, in general, are relatively common in older horses, but typically they are not functional. Schlotthauer [5] reported that 66% of horse necropsies had pathology of the thyroid glands, with benign adenomas being identified in 37% of cases. Additionally, Dalefield and Palmer [6] found thyroid masses in 11 of 29 horses necropsied, nine of which were horses older than 19 years. Typically, treatment is only pursued if the thyroid mass becomes locally invasive, leading to compression of the esophagus, trachea, lymphatics or vasculature, or if thyroid testing suggests a functional tumor.

Previous case studies of hyperthyroid equids have demonstrated clinical signs of weight loss, muscle wasting, polyphagia and tachycardia, but lack detailed description of any cardiac evaluation. Structural cardiac changes have been noted in humans [8], cats [9], geese [10], pigs [11], rabbits [12] and rats [13] with hyperthyroidism typically due to chronic hypertension and increased circulating blood volume. Tachycardia, tachyarrhythmias, high output heart failure, left ventricular hypertrophy and dilated cardiomyopathies are commonly reported signs in humans and cats [14,15,16]. Treatment of the hyperthyroid condition is recommended as the primary line of therapy, followed by management of cardiac conditions. Control of hyperthyroidism can lead to resolution of cardiac signs, but complete resolution varies based on the severity and progression of cardiac disease [14,16]. This case represents the first documentation of hemodynamically relevant cardiac remodeling in an equid associated with primary hyperthyroidism and describes the management and long-term follow up.

## 2. Materials and Methods

### 2.1. Initial Presentation

A 20-year-old mule gelding presented to Cornell University Equine and Nemo Farm Animal Hospital Internal Medicine Service for a three-month history of thin body condition, polyphagia, heart murmur and ventral cervical masses in the area of the throat latch. This mule was recently rescued and no history prior to his recent acquisition was available. Referring veterinarian bloodwork demonstrated elevated thyroid hormone, low-normal vitamin E concentration, mildly elevated selenium, mildly elevated resting adrenocorticotropic hormone (ACTH) measured in fall and negative Lyme and *Leptospira* antibody titers (see Table 1). The patient was referred for further work-up of possible primary hyperthyroidism.

On presentation, the mule was bright, alert and responsive, but anxious in the stall. He had a normal rectal temperature (99.7 F) and respiratory rate (24 breaths per minute). He was persistently tachycardic (60 beats per minute) and had a grade 5/6 pansystolic murmur heard loudest over the left cranial heart base. The heart sound S1 was not audible, S2 was heard consistently, while S3 and S4 were heard intermittently. Pulses were synchronous with moderate intensity. He weighed 325 kg with a body condition score of 2/9 and moderate to severe muscle wasting, most obvious over the top-line and gluteal region. Bilateral, asymmetric, firm masses were palpated in the ventral throat latch region of the neck, more prominent on the left side. Based on the location, the masses were suspected to be the thyroid glands. Additionally, there was clinically relevant dental disease with several missing premolar and molar teeth and severe wear of those remaining.

### 2.2. Pre-Operative Diagnostic Testing

Blood was collected for a full thyroid panel which showed marked elevation of all thyroid hormones (Table 1). The results of complete blood count and serum biochemistry were within the reference ranges except for a mildly elevated fibrinogen (600 mg/dL; RR = 0–200 mg/dL). An ionized calcium value was mildly elevated (1.78 mmol/L; RR = 1.4–1.72 mmol/L).

The throat latch masses were identified as thyroid glands using ultrasonography. The left thyroid was expanded by a well-encapsulated, well-vascularized mass consisting of loculated tissue of mixed echogenicity, measuring approximately 8 × 6 × 5 cm. The right thyroid gland contained homogenous, recognizable thyroid tissue in addition to hyperechoic nodules. It measured approximately 5 × 3.5 × 2.5 cm. The average size of a thyroid gland in an adult horse is 5 × 2.5 × 2.5 cm [16]. The top differential for the mule’s clinical signs and bloodwork abnormalities was primary hyperthyroidism.

Ultrasound of the thorax and abdomen was performed to assess for other concurrent disease processes that might contribute to the mule’s weight loss and were unremarkable. Additionally, endoscopy was performed of the upper airway to evaluate for metastasis of thyroid neoplasia; no abnormalities were visualized.

Due to the mule’s loud heart murmur and tachycardia, a cardiac evaluation was performed. Cardiac troponin I was normal (0 ng/mL, ref. 0–0.06 ng/mL). Non-invasive blood pressure (tail cuff, mmHg) was 172 systolic, 75 diastolic and 109 mean (based on an average of 3 measurements) (Reference: Systolic- 122 ± 20, Diastolic: 80 ± 17, Mean: 96 ± 19 mmHg) [18]. A 10 h resting electrocardiogram (Televet II, Heusenstamm, Germany) showed a persistent, mild tachycardia (median = 53 bpm; range = 18–166 bpm) as well as occasional atrial premature complexes (15/10 h) and infrequent ventricular premature complexes (2/10 h, Figure 1) in addition to a period of 2nd degree atrio-ventricular block, associated with sedation. An unsedated echocardiogram (GE Vivid E95 with 4VC probe) showed concentric hypertrophy of the left ventricle (increased relative and mean wall thickness with decreased left ventricular internal diameter), subjective right ventricular hypertrophy and increased left ventricular and left atrial systolic function (Table 2, Figure 2,) [18,19,20,21,22]. There was no evidence of valvular regurgitation, and the systolic murmur appeared to be the result of turbulent flow in the left ventricular outflow tract and aortic root during systole, although only suboptimal alignment with the flow was achieved to show mild flow acceleration on continuous wave Doppler imaging (Table 2, Figure 2).

### 2.3. Surgery and Treatments

Since the patient was in good systemic health and the cardiac disease appeared appropriately compensated, bilateral thyroidectomy was elected. The procedure was done under standing sedation using a detomidine constant rate infusion, at a rate of 0.1–0.5 mcg/kg/h. Local anesthesia was infiltrated subcutaneously around the border of the surgical field using mepivicaine. Sharp and blunt dissection were used to separate the skin and sternohyoid muscles of the ventral neck until the masses were identified. Both thyroid glands were well encapsulated. As the masses were gently pulled ventrally, the thyroid vasculature was identified and ligated. Upon inspection of the removed thyroid gland, a small flattened oval-shaped disk of yellow-brown tissue presumed to be a parathyroid gland was found adhered to the craniodorsal surface. The tissue was auto-transplanted into a pocket created in the left omohyoideus muscle. The subcutaneous layer and skin were closed routinely.

Pre and post-operative intravenous medications consisted of potassium penicillin (22,000 IU/kg IV q 6 h) for 48 h, gentamicin sulfate (6.6 mg/kg IV q 24 h) for 48 h and flunixin meglumine (1.1 mg/kg IV q 12 h) for 48 h. Omeprazole (3.1 mg/kg PO q 24 h) was started pre-op and continued for 10 days post-operatively. Trimethoprim/sulfamethoxazole (30 mg/kg PO q 12 h) was started two days post-operatively and continued for an additional seven days.

## 3. Results

### 3.1. Post-Operative Diagnostics

A thyroid panel was repeated approximately 24 h after the thyroidectomy and all values were markedly lower than pre-operative measures (see Table 2). A thyroid panel repeated three days after surgery, showed continuously dropping free T4 and total T4 and a similar T3 measurement.

Serum biochemistries were taken one, three and six days post-operatively revealed the phosphate had dropped below the reference range while calcium also decreased but remained within the range. By six days post-op, both calcium and phosphate were within their reference ranges again (Table 2). Ionized calcium was also assessed serially to monitor parathyroid function. Following surgery there was a mild decrease in ionized calcium that was still within the high end of the reference range. This value increased again above the reference range over the next few days, but by six days post-op was within the reference (Table 1).

### 3.2. Histopathology

The left thyroid gland was severely expanded by sheets and colloid-filled follicle-like structures of pleomorphic polygonal cells which exhibited moderate anisokaryosis and occasional mitotic activity. Features of malignancy were noted, including frequent capsular and lymphovascular invasion, consistent with a diagnosis of thyroid adenocarcinoma.

The right thyroid gland was expanded by multiple well-demarcated, encapsulated nodules of uniform thyroid follicular epithelial cells forming well-differentiated follicles, consistent with multiple thyroid adenomas. The stroma of the right thyroid gland was multifocally mineralized with areas of osseous metaplasia, considered an incidental finding in this case (Figure 3).

### 3.3. Follow Up

The mule was discharged three days post-operatively. A thyroid panel was repeated at a convenience interval by the referring veterinarian 17 days post-operatively which showed that all thyroid hormone concentrations were below the detectable range (Table 1). At this time the mule was started on 6 mg levothyroxine sodium orally, once daily (approximately 0.02 mg/kg), based on the low end of the recommended label dose. He began to gain weight and his polyphagia resolved but he did not achieve an ideal body condition score. The levothyroxine daily dose was increased first to 9 mg (0.03 mg/kg) which did not achieve total T4 within reference limits. After levothyroxine was increased to 18 mg (0.05 mg/kg), the patient’s T4 measured within reference range (Table 1). Despite normal total T4 achievement, the mule continued to have a low body condition score of 3/9, with a lack of topline muscle, 2 years post-operatively. Given the patient’s geriatric age, pituitary pars intermedia dysfunction was considered as an etiology for muscle wasting. Baseline ACTH was measured on EDTA plasma and found to be markedly elevated (Table 1). After initiating daily pergolide at a dose of 0.001 mg/kg orally, the patient’s ACTH concentration measured within reference range (25 pg/mL, ref. 2–30) and body condition score improved and muscle mass significantly improved.

Echocardiography performed one year post-operatively (GE Vivid iq with M5Ss probe) revealed persistently increased left ventricular relative wall thickness with similar left ventricular and left atrial systolic function as before surgery. At the time of this recheck the mule was tachycardic, however he was being treated for a severe lameness which may have confounded the evaluation. A grade 4/6 holosystolic left sided murmur over the left heart base was evident. Recheck at 2 years post-operative revealed a normal heart rate, improved left ventricular wall thickening, with normalized atrial and ventricular systolic function and smaller left atrial dimensions (Figure 4). The left sided systolic murmur was still evident but reduced in intensity (Grade 3/6, holosystolic, left heart base). No evidence of metastasis has been identified on physical examination 2 years post-operatively. Monitoring for metastasis has been recommended as well, along with serial blood thyroid concentration monitoring at least semi-annually so that levothyroxine supplementation can be titrated accordingly.

## 4. Discussion

Primary hyperthyroidism is a common disease process in cats, humans and other mammals. Primary hyperthyroidism in cats is most frequently caused by multinodular adenomatous hyperplasia; thyroid carcinomas account for less than 5% of clinical cases [23]. The most common causes in humans are Grave’s disease, toxic multinodular goiter, iodine-induced, adenomas or iatrogenic [24,25].

Although there are several common etiologies for primary hyperthyroidism in other species, there have been too few equid cases to uncover a common cause. Previous case reports of hyperthyroid horses have cited functional adenomas [2,4] or adenocarcinomas [1,3] as the source of excess thyroid hormone. Only one case reported involvement of both thyroid glands, one gland was diagnosed with a functional adenocarcinoma by histopathology while the other gland had signs of atrophy [3]. Previously used treatment options for hyperthyroidism are unilateral thyroidectomy [1,7], hemithyroidectomy [2] and propylthiouracil [3]. Extraction of the affected gland or a portion of the gland was effective in normalizing thyroid concentrations in these cases, but the animals in these cases had a remaining functional thyroid gland, unlike the current case study. The horse with bilateral thyroid involvement was treated with propylthiouracil orally which was successful in improving clinical signs and reducing thyroid hormone concentrations. Other treatment options discussed in this case included radioactive iodide ablation with I^131^ or medical management with methimazole or propylthiouracil, but these were not elected due to expense and lack of documented success [1,26]. Additionally, auto-transplantation of a parathyroid gland is commonly performed after thyroidectomy in humans and has been described in dogs, with most patients only demonstrating a transient hypoparathyroidism [27]. Mild sinus tachycardia was reported in all of these cases, but no murmurs, arrhythmias or other cardiac abnormalities were identified.

In general, excessive thyroid hormone has many effects on the cardiovascular system. First, excessive T3 directly and indirectly stimulates the renin-angiotensin-aldosterone (RAA) axis to increase circulating blood volume. Through several mechanisms, T3 results in relaxation of vascular smooth muscle cells resulting in peripheral vascular dilation and reduced vascular resistance. Reduced vascular resistance results in decreased renal perfusion, stimulating the release of renin which in turn promotes sodium and water reabsorption, increasing blood volume, in addition to the conversion of angiotensin I to angiontensin II which acts as a powerful vasoconstrictor. Additionally, thyroid hormone also directly stimulates erythropoietin production which can expand the red blood cell number. The initial decreased systemic vascular resistance reduces afterload and the increased blood volume results in a larger preload which ultimately increases stroke volume and cardiac output. The positive inotropic properties of T3, mediating expression of structural genes in cardiac myocytes can increase systolic function, further boosting stroke volume. Thyroid hormone’s manipulation of gene expression also results in positive chronotropic effects by altering the action potential duration and repolarization in cardiac pacemaker myocytes [15,28,29]. All of these changes result in increased cardiac output. Despite an initial decrease in peripheral vascular resistance, hyperthyroidism also results in increased arterial stiffness as the result of genomic and non-genomic factors targeting ion channels and endothelial nitric oxide synthesis [29]. The combination of arterial stiffness and increased cardiac output, combined with the activation of RAA axis eventually results in systemic hypertension and increased afterload. In this case, the measures of systolic function (fractional shortening and ejection fraction) were moderately increased on initial assessment, with evidence of concentric ventricular hypertrophy which slowly improved over the two-year follow up period.

In humans, chronic systolic hypertension is the leading cause of left ventricular concentric hypertrophy which is commonly the first cardiac architectural change detected in hyperthyroid humans [8] and was the primary architectural abnormality detected in this case. The left ventricle becomes hypertrophied as individual myocytes develop more sarcomeres to compensate for increased pressure without dilation of the ventricle itself. In contrast, from a pathophysiologic perspective, cats are more likely to develop eccentric hypertrophy when hyperthyroid [9]. Structural remodeling of the heart, such as hypertrophy of the left ventricle, can be reversed in many cases when the hypertension is controlled, but this process can take months to years. When the excess pressure is removed, interstitial fibrosis can begin to regress and the left ventricle becomes less hypertrophied [30]. Although much of the research on regression of hypertrophy is based on humans with aortic stenosis or insufficiency after aortic valve replacement, it is reasonable to extrapolate that the same would be true of humans or equids with chronic hypertension. In this case, follow up blood pressures are not available, so it is possible that the lack of full resolution of cardiac hypertrophy could be due to persistent chronic hypertension. The lack of full resolution in this case indicates investigation of the cardiac structures and function in hyperthyroid horses is warranted, particularly as left ventricular hypertrophy presents an important safety concern for the development of ventricular arrhythmias.

## 5. Conclusions

This is the first case report describing bilateral thyroidectomy of a functional thyroid tumor in a mule. Additionally, this is the first reported case of a hyperthyroid equid with structural and functional cardiac changes, similar to those seen in other mammals. Although thyroid tumors are common in older equids, there are very few that have been shown to be functional. Some clinical signs of hyperthyroidism may be misinterpreted as signs of aging (weight loss associated with poor dentition, early heart disease), resulting in under-diagnosis of primary hyperthyroidism in equids. Primary hyperthyroidism in equids is a rarely diagnosed condition and consideration should be given to the cardiovascular system when evaluating these cases.

## Figures and Tables

**Figure 1 animals-14-01660-f001:**
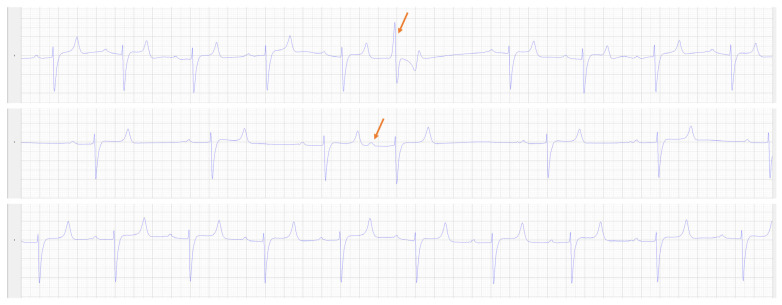
Modified base-apex ECG recordings from Lead II from the overnight ECG. The top panel shows a ventricular premature complex (orange arrow), the middle panel shows an atrial premature complex (orange arrow) while the bottom panel shows a period of sinus tachycardia (average rate 75 bpm).

**Figure 2 animals-14-01660-f002:**
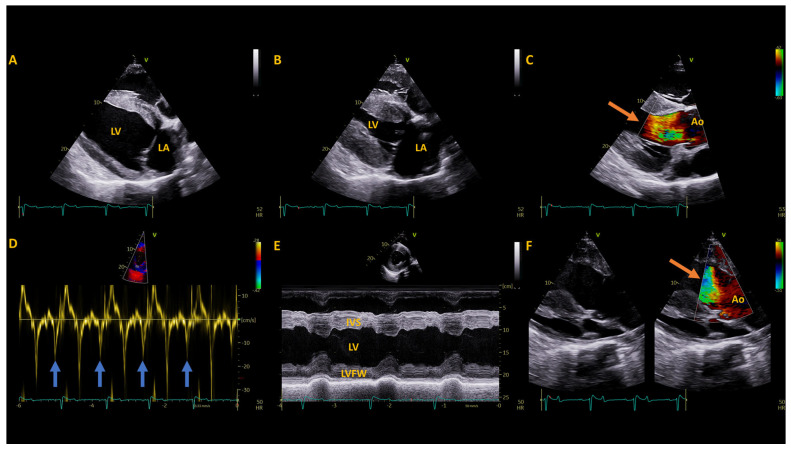
Echocardiographic images obtained at presentation. (**A**) A two-dimensional gray-scale right parasternal long axis view at the onset of systole, focused on the left atrium (LA) and left ventricle (LV). (**B**) A two-dimensional gray-scale right parasternal long axis view at end systole, focused on the LA and LV. The prominent LA and thickened ventricular walls can be appreciated. (**C**) A colour Doppler left ventricular outflow tract image at peak systole shows the turbulent flow (orange arrow) in the left ventricular outflow tract below the level of the aortic valve annulus. (**D**) A pulsed wave tissue Doppler image displaying the left ventricular free wall motion velocity from a right parasternal short axis view. The increased late diastolic wall motion velocity resulting from the atrial contribution to ventricular filling can be seen (blue arrows) following each P wave, resulting in the decreased E/A ratio (see Table 2). (**E**) An M-mode image of the left ventricle at the level of the chordae tendinae, indicating the left ventricular hypertrophy and decreased LV internal diameter resulting in the elevated relative wall thickness (see Table 2). (**F**) A two-dimensional gray-scale (left) and color Doppler (right) image of the left ventricular outflow tract from the left parasternal long axis view in peak systole. Flow acceleration can be seen as turbulent flow (orange arrow) in the left ventricular outflow tract just below the aortic valve annulus. Ao, aorta; IVS, interventricular septum; LVFW, left ventricular free wall.

**Figure 3 animals-14-01660-f003:**
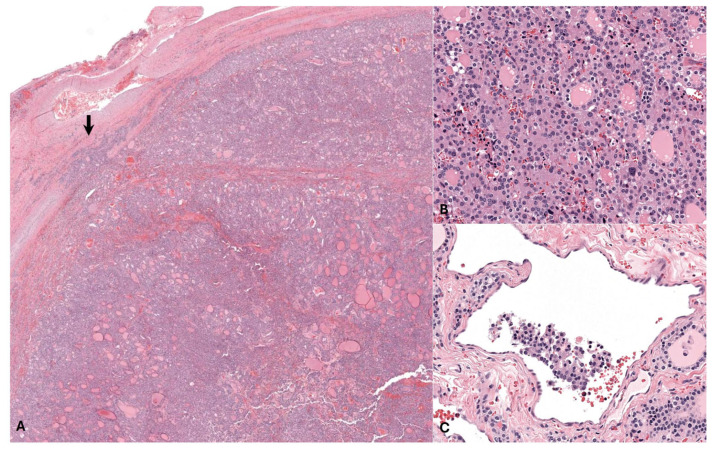
Photomicrographs of the left thyroid gland, hematoxylin and eosin (HE). (**A**) The thyroid gland is greatly expanded by a neoplasm of thyroid follicular epithelial cells with areas of capsular invasion (arrow). (**B**) The neoplastic cells often form colloid-filled follicle-like structures and exhibit moderate anisokaryosis with occasional karyomegaly and rare mitotic activity. (**C**) Blood and lymphatic vessels throughout the mass contain rafts of neoplastic cells, indicating vascular invasion, a feature of malignancy.

**Figure 4 animals-14-01660-f004:**
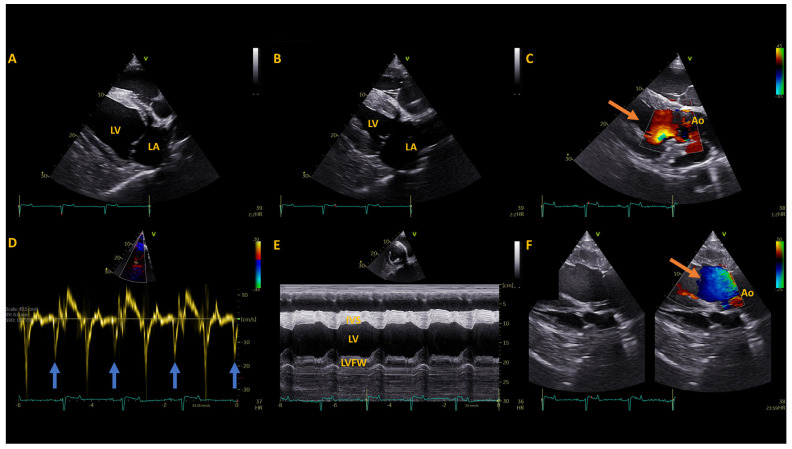
Echocardiographic images obtained two years after surgical removal of the functional thyroid tumor. (**A**) A two-dimensional gray-scale right parasternal long axis view at the onset of systole, focused on the left atrium (LA) and left ventricle (LV). (**B**) A two-dimensional gray-scale right parasternal long axis view at end systole, focused on the LA and LV. The less prominent LA and ventricular walls can be appreciated. (**C**) A colour Doppler left ventricular outflow tract image at peak systole shows reduced turbulent flow (orange arrow) in the left ventricular outflow tract below the level of the aortic valve annulus. (**D**) A pulsed wave tissue Doppler image displaying the left ventricular free wall motion velocity from a right parasternal short axis view. The increased late diastolic wall motion velocity resulting from the atrial contribution to ventricular filling can be seen (blue arrows) following each P wave, resulting in the decreased E/A ratio (see Table 2). (**E**) An M-mode image of the left ventricle at the level of the chordae tendinae, where a lesser degree of left ventricular hypertrophy with a similar LV internal diameter is observed, resulting in the reduced relative wall thickeness (see Table 2). (**F**) A two-dimensional gray-scale (left) and color Doppler (right) image of the left ventricular outflow tract from the left parasternal long axis view in peak systole. Less turbulent flow acceleration is seen (orange arrow) in the left ventricular outflow tract just below the aortic valve annulus. Ao, aorta; IVS, interventricular septum; LVFW, left ventricular free wall.

**Table 1 animals-14-01660-t001:** Pre and post-operative measurements of electrolytes and hormones [17]. Analyzers and equipment can be found in Appendix A.

Electrolytes (Reference)	Prior to Referral	Pre-op	12 h Post-op	24 h Post-op	2 Days Post-op	3 Days Post-op	6 Days Post-op	17 Days Post-op	3 Months Post-op	8 Months Post-op	10 Months Post-op	23 Months Post-op	25 Months Post-op
Calcium (10.9–12.9 mg/dL)		12.5		11.9		13.3	11.9						
Phosphate (2.1–4.7 mg/dL)		3.0		1.7		1.9	2.6						
Ionized Calcium (1.4–1.72 mmol/L)		1.78	1.65	1.68	1.8	1.81	1.68						
**Hormones (reference)**													
Total T4 (1–3 µg/dL)	5.78	6.7		3.4		0.45		<0.05	0.58	4.4	3.5	2.32	
Free T4 (1.2–1.8 ng/dL)	8.05	7.22		4.3		0.62		<0.15	0.38		2.05		
T3 (30–80 ng/dL)	204	289		43.8		43.5		<10	74.0		90.8		
Parathyroid hormone (1.3–15.0 mg/dL)						7.8							
Adrenocorticotropic Hormone (Fall: 30–90 pg/mL)	113											217	25
**Vitamins/minerals** **(reference)**													
Alpha-tocopherol (200–1000 µg/dL)	264												
Selenium (14–24 µg/dL)	25.34											20.08	

**Table 2 animals-14-01660-t002:** Cardiac measurements pre and post-operatively. **Bolded** measurements were considered abnormal for a 325 kg mule when compared to those reported for other equid species. References denoted by a * are mule-specific references [19,20]. All other provided values are extrapolated from equine references for a 500 kg horse [21,22].

Measurement (Units)	Available References (mean ± SD)	Pre-Operative	1 Year Post-Operative	2 Years Post-Operative
**Left Atrium**				
LAA_max_ (cm^2^)	92.8 ± 5.0	71.4	71.8	70.9
LAD_max_ (R) (cm)	11.9 ± 0.7	10	10.4	9.8
LA FAC active (%)	20 ± 7.0	**41**	**34**	**30**
LAD_max_ (L) (cm)	12.9 ± 0.5	**13.3**	**13.1**	11.2
LAA_max sx_ (cm^2^)	108.8 ± 12.2	**102.9**	87.4	74.6
LA/Ao (Sx) Ratio	2.5 ± 0.3	**2.7**	2.2	2.1
**Left Ventricle**				
IVS_d_ (cm)	2.8 ± 0.3 *	**3.2**	**3.3**	2.7
LVID_d_ (cm)	11.1 ± 1.5 *	**9**	**8.9**	**9.0**
LVFW_d_ (cm)	3.3 ± 0.3 *	2.6	2.6	2.8
IVS_s_ (cm)	4.4 ± 0.4	**5**	**4.7**	4.4
LVID_s_ (cm)	6.7 ± 1.3 *	**4.2**	**4.6**	**5.0**
LVFW_s_ (cm)	4.4 ± 0.4	4.3	3.9	3.9
FS (%)	40.4 ± 5.7 *	**54**	**49**	44
LVIV_d_ (ml)	1475 ± 200.7	694	672	757
LVIV_s_ (ml)	412 ± 81.6	146	148	189
SV (mL)	1065 ± 139.2	548	524	568
CO (L)	37.1 ± 6.02	30.6	30.4	21.2
EF (%)	71 ± 4.9	**79**	**78**	75
MWT (cm)	3.05 ± 0.2 *	2.9	2.9	2.8
RWT	0.52 ± 0.2 *	**0.65**	**0.66**	**0.61**
LAD/LVID ratio	1.1 ± 0.1	1.11	1.17	1.09
HR (bpm)	38 ± 4.4 *	**50**	**57**	37
**Great Vessels**				
PAD_ed_ (cm)	6.5 ± 0.47	5.8	5.8	5.9
AoD_ed_ (cm)	7.0 ± 0.9 *	6.6	6.7	6.5
PA_ed-sx_ (cm)	5.0 ± 0.31	4.1	4.1	4.7
AoA_sx_ (cm^2^)	45 ± 5.5	38	39.5	36
Ao/PA (sx) Ratio	1.4 ± 0.1	1.6	1.6	1.4
**Tissue Doppler Imaging**				
IMP		0.38	0.29	0.44
Em/Am	3.1 ± 0.8	**1.7**	**1.2**	**1.9**
**Velocity**				
AoV_max_ (m/s)		**1.1**	**1.2**	0.9

LAA_max_—Left atrial area at maximum size; LAD_max_ (R)—Maximal left atrial diameter from the right; LA FAC active—Left atrial fractional area change—active; LAD_max_ (L)—Maximal left atrial diameter from the left; LAA_max sx_—maximal left atrial area—short axis; LA/Ao (sx)—left atrial/aorta area (short axis) ratio; IVS_d_—interventricular septum thickness in diastole; LVID_d_—left ventricular internal diameter in diastole; LVFW_d_—left ventricular free wall thickness in diastole; IVS_s_—interventricular septum thickness in systole; LVID_s_—left ventricular internal diameter in systole; LVFW_s_—left ventricular free wall thickness in systole; FS—Fractional shortening; LVIV_d_—left ventricular internal volume in diastole; LVIV_s_—left ventricular internal volume in systole; SV—Stroke volume; CO—cardiac output; EF—Ejection fraction; MWT—mean wall thickness; RWT—relative wall thickness; LAD/LVID ratio—left atrial diameter from the right to left ventricular internal diameter in diastole ratio; PAD_ed_—pulmonary artery diameter at end diastole; AoD—Aortic annulus diameter at end diastole; PA_sx_—pulmonary artery diameter in short axis at end diastole; AoA_sx_—aortic area in short axis; Ao/PA (sx)—aorta and pulmonary artery in short axis ratio; IMP—index of myocardial performance; Em/Am—left ventricular free wall velocity in early/late diastole ratio; AoV_max_—maximal aortic outflow velocity.

## Data Availability

The original contributions presented in the study are included in the article, further inquiries can be directed to the corresponding authors.

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
