# Peer review of "Cardiac Disease Related to Primary Hyperthyroidism in a 20-Year-Old Mule"

_animals, 2024, doi:10.3390/ani14111660_

Round 1
Reviewer 1 Report
Comments and Suggestions for Authors
It's a great case and is well reported here. You can usefully tighten it up a bit by improving the English grammar and avoiding some repetitions.
We know that thyroid disorders of the horse are very poorly reported and are also very poorly managed. As we build up information regarding genuine documented cases, we will start to get more information about how we can dispel the misconceptions regarding the thyroid gland of the horse. I am pleased that you have set out to try to clarify the case in such great detail and I view that as a really strong positive component of this paper.Since this is probably the first case of fully explored and documented hyperthyroidism in horses, it should go some way toward confirming the existence at least of hyperthyroidism. As such, I hope it will prove to be really interesting for both clinicians at practice and referral centres and researchers.
Many of the previous reported effects of over or under activity of the thyroid gland of the horse are poorly documented and do not contain adequate information to provide useful long-term support of material.
Well done for writing the paper so well – it is well constructed and whilst the concepts are often difficult to interpret adequately in the face of poor general knowledge in the subject in equine species, you have done well to create a circumstance that I believe would be attractive to a reader and will contribute to further improvement in our understanding of the diseases associated with this structure. I was particularly interested in the restoration of the horse to a more or less normal state following the medication supplementation – particularly given the intimate relationship between the thyroid and the parathyroid in the horse.
Case descriptions are challenging in this respect because they are single events at where the disease is extremely rare like this, it is important that we make the maximum use of them and the outcome is worth assessing in the long term as well as in the relative short-term.
Comments on the Quality of English Language
The language is good - a few grammar corrections, but otherwise, it's a really good paper. Well done
Author Response
Reviewer #1:
Response: Thank you for your comments and contributions. We have corrected grammatical errors and tried to remove any redundancies throughout.
Reviewer 2 Report
Comments and Suggestions for Authors
The case presented in the article titled "Cardiac disease associated with primary hyperthyroidism in a 20-year-old Mule" is very interesting and the text is well-written. However, I have a few comments that I think would further improve the quality of the article.
In line 75, there is a test for Lyme disease and Leptospira, but nowhere do I see a justification for selecting only these diseases - please explain.
The description lacks photos from comparative ultrasound images - the same projections during the control, showing changes in the image. Please add.
There are also no photos from the ECG recording. Please add.
Some test results (such as blood pressure measurement) lack information about reference values ​​and comments on whether the result is within or outside the norm.
Again, no photo from histological examination of the thyroid gland. Please add.
I'm also wondering about the control time - line 199 - day 17. Why then? Please explain.
Please justify the selection of the dose of levothyroxine sodium - was the dose taken from horses or from humans? Please add.
In the summary of the patient's description, there is no clear information on further treatment and recommendations. Please add.
Author Response
Reviewer #2
The case presented in the article titled "Cardiac disease associated with primary hyperthyroidism in a 20-year-old Mule" is very interesting and the text is well-written. However, I have a few comments that I think would further improve the quality of the article.
In line 75, there is a test for Lyme disease and Leptospira, but nowhere do I see a justification for selecting only these diseases - please explain.
Response: These values were run by the referring veterinarian prior to presentation and diagnosis of hyperthyroidism. They were included for completeness of history.
The description lacks photos from comparative ultrasound images - the same projections during the control, showing changes in the image. Please add.
Response: The same echocardiographic views taken at the 2yr recheck have been added as Figure 4.
There are also no photos from the ECG recording. Please add.
Response: Figure 1, showing representative arrhythmias and sinus tachycardia is included.
Some test results (such as blood pressure measurement) lack information about reference values ​​and comments on whether the result is within or outside the norm.
Response: Reference intervals are not available in mules specifically, but references for horses were added to the text and references throughout text adjusted accordingly.
Added: “(Reference: Systolic- 122 ± 20, Diastolic: 80 ± 17, Mean: 96 ± 19 mmHg) [18].”
Reference added: Heliczer N, Lorello O, Casoni D, et al. Accuracy and Precision of Noninvasive Blood Pressure in Normo-, Hyper-, and Hypotensive Standing and Anesthetized Adult Horses. J Vet Intern Med 2016;30:866-872.
Again, no photo from histological examination of the thyroid gland. Please add.
Response: Histologic images (Figure 3) has been added to the manuscript, and the histologic text has been updated to reflect this.
The following has been added to the text:
“Histopathology: The left thyroid gland was severely expanded by sheets and colloid-filled follicle-like structures of pleomorphic polygonal cells which exhibited moderate anisokaryosis and occasional mitotic activity. Features of malignancy were noted, including frequent capsular and lymphovascular invasion, consistent with a diagnosis of thyroid adenocarcinoma.
The right thyroid gland was expanded by multiple well-demarcated, encapsulated nodules of uniform thyroid follicular epithelial cells forming well-differentiated follicles, consistent with multiple thyroid adenomas. The stroma of the right thyroid gland was multifocally mineralized with areas of osseous metaplasia, considered an incidental finding in this case. (Figure 3).”
“Figure 3. Photomicrographs of the left thyroid gland, hematoxylin and eosin (HE). A) The thyroid gland is greatly expanded by a neoplasm of thyroid follicular epithelial cells with areas of capsular invasion (arrow). B) The neoplastic cells often form colloid-filled follicle-like structures and exhibit moderate anisokaryosis with occasional karyomegaly and rare mitotic activity. C) Blood and lymphatic vessels throughout the mass contain rafts of neoplastic cells, indicating vascular invasion, a feature of malignancy.”
I'm also wondering about the control time - line 199 - day 17. Why then? Please explain.
Response: Day 17 was simple a recheck interval of convenience based on owner and RDVM availability. The text has been updated to reflect this.
Added: “A thyroid panel was repeated at a convenience interval by the referring veterinarian 17 days post-operatively…..”
Please justify the selection of the dose of levothyroxine sodium - was the dose taken from horses or from humans? Please add.
Response: The dose was started per the Thyro-L dosing label, at the low end of the range.
The following was added to the text:
Line 230: “….based on the low end of the recommended label dose.”
In the summary of the patient's description, there is no clear information on further treatment and recommendations. Please add.
Response: The following was added to the text to the abstract
Line 35: “Continued bi-annual follow up and monitoring of thyroid levels was recommended.”
Reviewer 3 Report
Comments and Suggestions for Authors
The work deals with a rarely described issue in mules in particular, about which work in general is sporadic.
The introduction is comprehensive and also includes references to other animal species, including man.
2.2. please specify which apparatus was used for blood tests
please indicate which ultrasound machine was used and which probe
3.2. please specify whether only histopathological HE examination was performed or perhaps some other examination
Discussion is appropriate.
References are appropriate
Author Response
Reviewer #3
The work deals with a rarely described issue in mules in particular, about which work in general is sporadic.
The introduction is comprehensive and also includes references to other animal species, including man.
2.2. please specify which apparatus was used for blood tests
Response: If the editors feel these details are necessary, they have been added to the text – lines 360-369.
- Calcium and phosphate were measured using the Cobas Roche chemistry analyzer
- Ionized calcium was measured using Zoetis iSTAT
Hormones were measured using the following:
- Parathyroid – Immunodiagnostic Systems
- ACTH, Total T3 and Total T4 – Immulite 2000 CIA, Siemens Healthineers,
- Total T3 – Antech RIA
- Alpha tocopherol – high performance liquid chromatography with fluorescence detector (HPLC-FLD) methodology
- Selenium – Graphite furnas atomic absorption spectrometry (AAS) methodology
please indicate which ultrasound machine was used and which probe
Response: Images were acquired with GE Vivid E95 with 4VC probe and the Vivid iq with the M5Sc probe respectively.
The following was added to the text, line 122: “….(Vivid E95 with 4VC probe)…” and line 243: “… Vivid iq with M5Sc probe)…”
3.2. please specify whether only histopathological HE examination was performed or perhaps some other examination
Response: Histologic images (figure 3) has been added to the manuscript, and the histologic text has been updated to reflect this.
The following has been added to the text:
“Histopathology: The left thyroid gland was severely expanded by sheets and colloid-filled follicle-like structures of pleomorphic polygonal cells which exhibited moderate anisokaryosis and occasional mitotic activity. Features of malignancy were noted, including frequent capsular and lymphovascular invasion, consistent with a diagnosis of thyroid adenocarcinoma.
The right thyroid gland was expanded by multiple well-demarcated, encapsulated nodules of uniform thyroid follicular epithelial cells forming well-differentiated follicles, consistent with multiple thyroid adenomas. The stroma of the right thyroid gland was multifocally mineralized with areas of osseous metaplasia, considered an incidental finding in this case. (Figure 3).”
“Figure 3. Photomicrographs of the left thyroid gland, hematoxylin and eosin (HE). A) The thyroid gland is greatly expanded by a neoplasm of thyroid follicular epithelial cells with areas of capsular invasion (arrow). B) The neoplastic cells often form colloid-filled follicle-like structures and exhibit moderate anisokaryosis with occasional karyomegaly and rare mitotic activity. C) Blood and lymphatic vessels throughout the mass contain rafts of neoplastic cells, indicating vascular invasion, a feature of malignancy.”
Discussion is appropriate.
References are appropriate
Round 2
Reviewer 3 Report
Comments and Suggestions for Authors
The paper may be published